# Transformer-based out-of-distribution detection for clinically safe segmentation

**Mark S. Graham**[1]                                       MARK.GRAHAM@KCL.AC.UK
**Petru-Daniel Tudosiu**[1]                                PETRU.TUDOSIU@KCL.AC.UK
**Paul Wright**[1]                                               P.WRIGHT@KCL.AC.UK
**Walter Hugo Lopez Pinaya**[1]                     WALTER.DIAZ_SANZ@KCL.AC.UK
**Jean-Marie U-King-Im**[2]                              JU-KING-IM@NHS.NET
**Yee H. Mah**[1,2]                                            YEE.MAH@NHS.NET
**James T. Teo**[2,3]                                          JAMESTEO@NHS.NET
**Rolf Jager**[4]                                                 R.JAGER@UCL.AC.UK
**David Werring**[5]                                           D.WERRING@UCL.AC.UK
**Parashkev Nachev**[4]                                     P.NACHEV@UCL.AC.UK
**Sebastien Ourselin**[1]                                   SEBASTIEN.OURSELIN@KCL.AC.UK
**M. Jorge Cardoso**[1]                                      M.JORGE.CARDOSO@KCL.AC.UK

[1] *Department of Biomedical Engineering, School of Biomedical Engineering & Imaging Sciences, King's College London, London, UK*

[2] *King's College Hospital NHS Foundation Trust, Denmark Hill, London, UK*

[3] *Institute of Psychiatry, Psychology & Neuroscience, King's College London, London, UK*

[4] *Institute of Neurology, University College London, London, UK*

[5] *Stroke Research Centre, UCL Queen Square Institute of Neurology, London, UK*

**Editors:** Under Review for MIDL 2022

## Abstract

In a clinical setting it is essential that deployed image processing systems are robust to the full range of inputs they might encounter and, in particular, do not make confidently wrong predictions. The most popular approach to safe processing is to train networks that can provide a measure of their uncertainty, but these tend to fail for inputs that are far outside the training data distribution. Recently, generative modelling approaches have been proposed as an alternative; these can quantify the likelihood of a data sample explicitly, filtering out any out-of-distribution (OOD) samples before further processing is performed. In this work, we focus on image segmentation and evaluate several approaches to network uncertainty in the far-OOD and near-OOD cases for the task of segmenting haemorrhages in head CTs. We find all of these approaches are unsuitable for safe segmentation as they provide confidently wrong predictions when operating OOD. We propose performing full 3D OOD detection using a VQ-GAN to provide a compressed latent representation of the image and a transformer to estimate the data likelihood. Our approach successfully identifies images in both the far- and near-OOD cases. We find a strong relationship between image likelihood and the quality of a model's segmentation, making this approach viable for filtering images unsuitable for segmentation. To our knowledge, this is the first time transformers have been applied to perform OOD detection on 3D image data.

**Keywords:** Transformers, out-of-distribution detection, segmentation, uncertainty.

## 1. Introduction

An important aim of the medical deep learning field is to develop image processing algorithms that can be deployed in clinical settings. These tools need to be robust to the full range of potential inputs they might receive in a clinical context. We can expect this data to be much more diverse than the typically clean, sanitised datasets on which these algorithms are developed and tested. Even if attempts are made to train on messier datasets with images exhibiting artefacts and other issues, we would expect that eventually the tool will be presented with data it has not seen during training. Typically, deep learning networks perform well when operating in-distribution, but performance can degrade unpredictably and substantially when operating on data out-of-distribution (OOD).

One approach to this problem is to develop networks that provide both predictions and a measure of their uncertainty, enabling decisions to be referred to humans when they are presented with difficult or OOD data samples. Bayesian Neural Networks (BNN) that learn a distribution of weights are capable of this; one popular aproach is to appromixate a BNN using dropout-based variational inference (Gal and Ghahramani, 2016). Another common approach is the deployment of an ensemble of neural networks (Lakshminarayanan et al., 2017). A comprehensive evaluation of uncertainty methods for classification found that the quality of uncertainty measures degraded as the size of the distributional shift increased (Ovadia et al., 2019). Some work has evaluated these methods in the context of image segmentation but is typically confined to evaluating the utility of the uncertainty methods in-distribution (Jungo et al., 2018; Nair et al., 2020); work that does evaluate the uncertainty in an OOD setting typically investigates only small dataset shifts such as increased noise (Haas and Rabus, 2021) or lower quality scans (McClure et al., 2019).

A second approach is to filter our anomalous data before it is fed to the task-specific network by using a generative model that can quantify the probability that a data sample is drawn from the distribution that the task-specific model was trained on. Of the generative approaches, autoregressive methods are attractive for two reasons: firstly, they allow for the computation of exact likelihoods, and secondly, a class of architectures that can be used for autoregressive prediction termed transformers (Vaswani et al., 2017) are proving highly effective general-purpose architectures, achieving state-of-the-art performance across a range of tasks in language (Devlin et al., 2018; Brown et al., 2020) and, increasingly, vision (Dosovitskiy et al., 2020). The high dimensionality of medical images makes it computationally infeasible to use transformers to model the sequence of raw pixel values, and a recent body of work has instead used transformers to model the compressed discrete latent space of an image obtained from a vector-quantised network such as a VQ-VAE or VQ-GAN (Oord et al., 2017; Razavi et al., 2019; Esser et al., 2021). This approach has provided state-of-the-art unsupervised pathology segmentation for 2D medical images (Pinaya et al., 2021). Prior work has also shown that this vector-quantised class of auto-encoding methods can be used to substantially compress 3D medical images (Tudosiu et al., 2020) indicating the VQ-GAN + transformer approach might be applied to fully 3D anomaly detection but, to our knowledge, there is no published work attempting this.

In this work, we make two principal contributions, focusing on the problem of segmentation of haemorrhagic lesions in head CT data. Firstly, we evaluate segmentation uncertainty methods on a range of OOD inputs and demonstrate they can catastrophically fail, produc-

ing confidently wrong predictions. Secondly, we use transformers to perform image-wide OOD detection on 3D images. We find this can effectively flag OOD data that segmentation networks fail to perform well on, demonstrating their viability as a filter in clinical settings where robust and fully-automated segmentation pipelines are needed.

## 2. Methods

In this work, we focus on the challenge of segmenting Intracerebral Haemorrhages (ICH) in head CT data. The following sections summarise the datasets used, the trained segmentation networks, and the approach to training the VQ-GANs and transformers.

### 2.1. Datasets

We use three main datasets in this work; two head CT datasets (one used for training, and an independent one for model evaluation), and a non-head CT dataset.

The **CROMIS** dataset is a set of 687 head CT scans used for training all the networks in this paper. The data consists of CTs containing ICH, acquired across multiple sites as part of a trial (Werring, 2017; Wilson et al., 2018). Ground-truth haemorrhage segmentation masks are available for 221 scans in the dataset.

The **KCH** dataset was used for algorithm validation. It consists of 47 clinical scans selected for the presence of ICH, all with ground-truth masks provided by an experienced neuroradiologist. This dataset was used to represent in-distribution test data; it was further used to produce a set of corrupted scans to test our algorithms in the near-OOD setting. A range of corruptions were applied to each scan in the dataset, designed to emulate a number of scenarios such as imaging artefact, image header errors, and errors in the preprocessing pipeline that is typically applied before data is input into a network. The corruptions included: addition of Gaussian noise, inversion through each of the three image planes, skull-stripping, setting the image background to values not equal to 0, global scaling of all image intensities by a fixed factor, and the deletion of a set of slices (or chunks) of the image. Any spatial manipulations of the image were also applied to the labels. This totalled 15 corruptions applied to each image, creating a corrupted dataset of 705 images. Examples of the corruptions applied are included in Appendix A.

The **Medical Decathlon** dataset was used to test our algorithms in the far-OOD setting. It comprises of a number of medical images covering a variety of organs and imaging modalities, none of which are head CT. We selected 22 images from the test set of each of the ten classes (or as many as were available in the test set if less than 22). A more detailed description of this dataset can be found in (Simpson et al., 2019).

Data processing was harmonised between all datasets as much as possible. All CT head images were registered to MNI using an affine transformation, resampled to 1mm isotropic, tightly cropped to a $176 \times 208 \times 176$ grid, intensities clamped between $[-15, 100]$ and then rescaled to the range $[0, 1]$. For the images in the Decathlon dataset, all were resampled to be 1mm isotropic and either cropped or zero-padded depending on size to produce a $176 \times 208 \times 176$ grid. All CT images had their intensities clamped between $[-15, 100]$ and then rescaled to lie in the range $[0, 1]$, all non-CT images were rescaled based on their minimum and maximum values to lie in the range $[0, 1]$.

## 2.2. Segmentation networks

We tested three uncertainty methods commonly employed in the literature. The first is intended as a simple baseline and uses the softmax of the network's output as a per-pixel probability map. The second is an ensemble of $N$ neural networks, identical in architecture but each trained on a different subset of the data (Lakshminarayanan et al., 2017). Based on recommendations from (Ovadia et al., 2019) we chose $N$=5. Finally, we use an approximation of a BNN obtained through dropout-based variational inference, training each dropout layer with a dropout probability of $p = 0.5$ and using 5 passes during inference to approximate the posterior (Gal and Ghahramani, 2016).

All networks used the same UNet backbone based on (Falk et al., 2019) and implemented in Project MONAI[1] as the 'BasicUnet' class, with (32, 32, 64, 128, 256) features in the 5 encoding layers, instance normalisation, and LeakyReLU activations. We trained the networks using a batch size of 3, augmenting with affine and elastic transformations and sampling patches of size $128^3$. The Dice loss was used except for the baseline network, which was trained using cross-entropy loss as Dice is known to provide poorly calibrated, overly-confident predictions (Mehrtash et al., 2020). All networks were optimised using the AMSGrad variant of Adam (Reddi et al., 2019) with a learning rate of $1e^{-3}$.

For each network we sought to assign a single uncertainty value to each predicted lesion. Firstly, we produced a per-voxel uncertainty map for each method. For the baseline method this was the per-voxel softmax of the network output. For the remaining networks we used the entropy between the $N$ predictions at each voxel, $\left(1 - \sum_i^N p_i \ln p_i\right)$ as described in (Nair et al., 2020), subtracting the entropy from 1 to produce a measure where larger values reflect higher certainty. We produce per-lesion certainty by taking the average of the per-voxel measures across each lesion, where each separate lesion is taken as each fully connected component from the majority vote prediction of each network.

## 2.3. VQ-GAN + Transformer networks

Our approach to outlier detection uses a VQ-GAN to compress the information content of each 3D volume into a discrete latent representation, and a transformer to learn the probability density of these representations.

The VQ-GAN (Esser et al., 2021) contains an encoder $E$ which takes input $x \in \mathbb{R}^{H \times W \times D}$ and produces a latent representation $z \in \mathbb{R}^{h \times w \times d \times n}$ where $n$ is the dimension of the latent embedding vector. The representation is quantised by finding its nearest neighbour, as measured by an $L_2$ norm, in a codebook of $K$ $n$-dimensional vectors and replacing the representation with the nearest neighbour's codebook index, $k$. A decoder uses the quantised latent space to reconstruct the input, $\hat{x} \in \mathbb{R}^{H \times W \times D}$. To encourage the network to learn a rich codebook, a discriminator $D$ is used to try to differentiate between real and reconstructed images. Our implementation's encoder contains four levels, each consisting of a convolution with stride=2 and a residual layer, each followed by ReLU layers. This produces a latent space $16\times$ smaller along each dimension, so an input with size $176 \times 208 \times 176$ is compressed to a latent size of $11 \times 13 \times 11 = 1573$ elements. The codebook has $K = 256$, each with dimension $n = 256$. The decoder also contained four levels, each consisting of

---

1. https://monai.io/

a residual layer followed by a transposed convolution with stride=2. The codebook was updated using the exponential moving average as described in (Oord et al., 2017). The VQ-GAN paper combined a mean-squared error loss and a perceptual loss (Zhang et al., 2018) for the reconstruction loss - we used both these and an additional spectral loss (Dhariwal et al., 2020). Given state-of-the-art anomaly detection results have been reported in 2D using a simpler VQ-VAE with MSE loss (Pinaya et al., 2021), we also performed an ablation study to understand how the additional components of the VQ-GAN contributed to performance. Models were trained using Adam with a learning rate=$1.65e^{-4}$ and a batch-size of 96 on a Nvidia DGX A100.

After training the VQ-GAN, we can estimate the probability density of the training data using a transformer. Each 3D discrete representation obtained from the trained VQ-GAN is flattened into a 1D sequence, and the data-likelihood is represented as the product of conditional probabilities, $p(x) = \prod_i^n p(x_i|\mathbf{x}_{<i})$, with the transformer learning the distribution of $p(x_i|\mathbf{x}_{<i})$ by being trained to maximise the log-likelihood of the training data. In addition to estimating the whole image likelihood $p(x)$, we produced spatial likelihood maps by reshaping each $p(x_i|\mathbf{x}_{<i})$ from the 1D sequence back into the 3D shape of the latent representation and upsampling to produce a spatial likelihood map of the same dimension as the input image. The transformer's attention mechanism has a quadratic memory dependence on sequence length that makes it difficult to train on large sequences, so we made use of the more efficient Performer architecture (Choromanski et al., 2020) which uses a linearised approximation of the attention matrices to allow for training on longer sequences. We used a 22 layer Performer with 8 attention heads and a latent representation of size 256. The model was trained using the cross-entropy loss using a learning rate of $5e^{-4}$ and a batch size of 240 on a Nvidia DGX A100.

Both models were trained on the full CROMIS dataset. It should be noted that this dataset contains pathological images containing haemorrhages; the definition of in-distribution here is not healthy scans but rather scans that are similar to the segmentation network's training set, and the aim is to estimate whether a new input is similar enough to the segmentation network's training set, so that it will be segmented accurately.

## 3. Experiments and Results

### 3.1. Segmentation uncertainty

We firstly examine the performance of segmentation algorithms in the far-OOD case where images are of a different organ and/or modality than the intended target for segmentation. In this case, any detection can be considered a false-positive (FP). We calculated per-lesion confidence scores for each detection and compared them to the per-lesion scores of every true-positive (TP) detection on the normal head CT dataset. Figure 1 shows the distribution of lesion confidence scores for these two datasets overlap regardless of the segmentation uncertainty method used, meaning it is not possible to separate FP detections made on far-OOD data from TP detections on in-distribution data using any of these lesion confidence scores alone. This motivates the need for explicit OOD detection models.

Secondly, we look at the segmentation network's uncertainty performance for near-OOD data: corrupted head CT scans. As these scans contain haemorrhages both TPs and FPs exist. We defined a TP as a predicted mask with at least 50% overlap with a ground truth

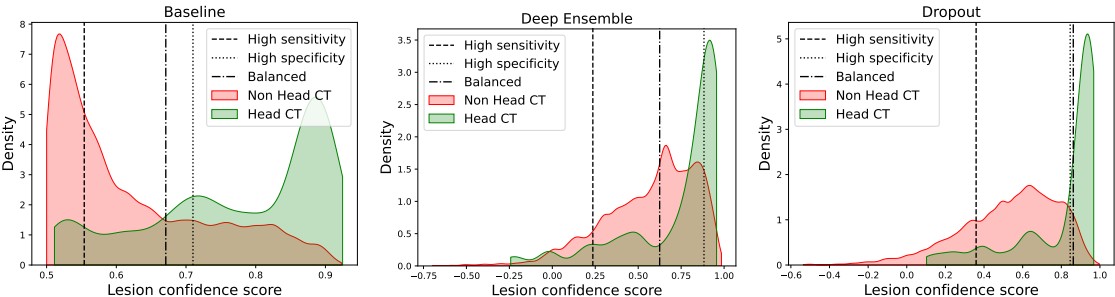

Figure 1: The distribution of per-lesion confidence scores for far-OOD non-head CT data and all true positive detections in the head CT dataset. Lines also show confidence thresholds for operating in three regimes: high-sensitivity ($> 90\%$), high-specificity ($> 90\%$) and balanced, as determined using each method on the head CT dataset.

mask, and computed the AUC obtained when using the per-lesion certainty scores to classify detections, see Table B.1. The networks are able to classify lesions relatively well for certain types of corruption, including noise, image flipping, and the removal of 'chunks' from the data. However, they perform poorly for images with modified background values or those that have been skull-stripped. For scaled images all three methods had an AUC $\leq 0.36$, showing they often assigned higher confidence values to FP detections than TP detections.

### 3.2. Transformers for OOD detection

We examined the ability of transformers to filter out OOD inputs based on the whole-image log-likelihood. Figure 2 shows the distribution of log-likelihood values for far-OOD, near-OOD and in-distribution classes (plots showing each sub-class can be found in Figure B.1). Table 1 show the ability of the log-likelihood to distinguish OOD classes from normal head CT data. Performance is perfect for the far-OOD case. In the case of near-OOD data, classes on which the segmentation uncertainty performed poorly are distinguished well: images with adapted backgrounds, skull-stripping, and global intensity scaling are all distinguished with an AUC=1. Subtler corruptions, namely noise with $\sigma \leq 0.1$ and L-R flips, are not distinguished well. These are classes for which the segmentation network uncertainty measures perform well, suggesting these corruptions are more in-distribution and explaining why they have been assigned likelihoods more similar to the normal head CT data. This result also suggests transformers and segmentation networks with uncertainty may be used in tandem, with highly OOD images being filtered out by the transformer and the segmentation network providing meaningful uncertainty estimates on images that are only slightly OOD. Figure B.2 shows some qualitative results on real data: the CROMIS volumes with the lowest and highest log-likelihood values as assigned by the transformer. Figure B.3 shows that reconstruction MSE alone is unable to separate out in-distribution and OOD data, indicating the transformer component is essential for OOD detection.

Table B.2 reports an ablation study for these results. The results demonstrate the VQ-GAN outperforms the VQ-VAE; but that both the perceptual loss and adversarial loss are needed together to improve performance. Our inclusion of an additional spectral loss

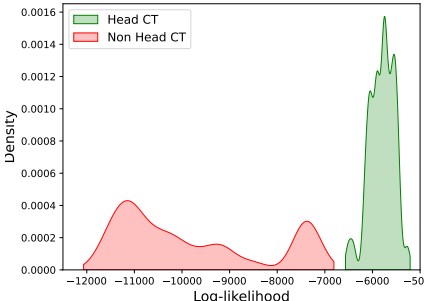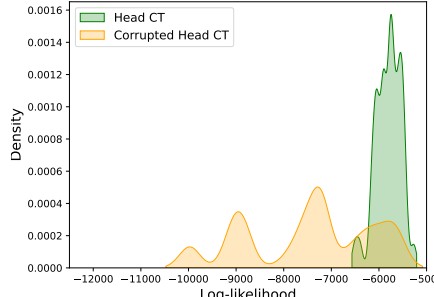

Figure 2: Data log-likelihoods for far-OOD, near-OOD and in-distribution volumes using transformers. Log-likelihoods for each sub-class are included in Figure B.1.

provides a modest improvement in performance over the standard VQ-GAN losses. Examination of spatial likelihood maps in Figure B.4 show good localisation of image corruptions. These maps might be used to explain what part of the image has caused the system to flag it as low-likelihood. A common failure mode reported for generative approaches to anomaly detection is the assignment of higher log-likelihoods to very OOD samples (Nalisnick et al., 2018); this did not occur in our experiments, which suggests that transformer-based anomaly detection methods could be more robust to this type of failure.

We further sought to characterise the relationship between an image's likelihood and the performance of the segmentation network; we show results for the best-performing dropout network in Figure 3 and include results for other networks, which are similar, in Figure B.5. While we would not necessarily expect a relationship between examples the transformer assigns a low likelihood to and examples the networks perform poorly on, our results indicate there is a strong one and that poor segmentations could be effectively filtered out using a log-likelihood threshold of -7000. Furthermore, the results show that corrupted CT-scans that were assigned similar likelihoods to the in-distribution data tend to be handled better by the segmentation network, supporting our claim that these indeed are more subtle corruptions.

## 4. Conclusion

In a clinical setting, deployed systems must be robust to the full range of data they may be presented with. Our results show that commonly used methods for obtaining uncertainty from segmentation networks fail when OOD, making confidently wrong predictions in far-OOD and near-OOD settings. We demonstrate that with a VQ-GAN for image compression and a transformer for density estimation we can successfully detect both far- and near-OOD data, and we find the images flagged as OOD by this approach are precisely the ones that segmentation networks struggle with. These results suggest our approach can be used to filter out OOD inputs that a segmentation network is likely to fail on, ensuring it is only run on in-distribution data they are likely to perform well on. Future work will focus on further validating these findings on real-world clinical datasets.

Table 1: AUC for distinguishing between normal head CT and far- and near-OOD classes.

| | Dataset | Likelihood | AUC |
|---|---|---|---|
| Non Head CT | Head MR | -7288 (134) | 1.00 |
| | Colon CT | -10809 (789) | 1.00 |
| | Hepatic CT | -10712 (763) | 1.00 |
| | Hippocampal MR | -7465 (20) | 1.00 |
| | Liver CT | -11116 (658) | 1.00 |
| | Lung CT | -9957 (289) | 1.00 |
| | Pancreas CT | -10798 (791) | 1.00 |
| | Prostate MR | -9140 (134) | 1.00 |
| | Spleen CT | -10895 (382) | 1.00 |
| | Cardiac MR | -9661 (318) | 1.00 |
| Corrupted Head CT | Noise $\sigma = 0.01$ | -5796 (253) | 0.49 |
| | Noise $\sigma = 0.1$ | -5793 (237) | 0.49 |
| | Noise $\sigma = 0.2$ | -6637 (324) | 0.98 |
| | BG value=0.3 | -9022 (89) | 1.00 |
| | BG value=0.6 | -8803 (100) | 1.00 |
| | BG value=1.0 | -9979 (127) | 1.00 |
| | Flip L-R | -5850 (253) | 0.55 |
| | Flip A-P | -7435 (205) | 1.00 |
| | Flip I-S | -9036 (165) | 1.00 |
| | Chunk top | -6382 (214) | 0.96 |
| | Chunk middle | -7784 (179) | 1.00 |
| | Skull stripped | -7226 (125) | 1.00 |
| | Scaling 10% | -7436 (119) | 1.00 |
| | Scaling 1% | -7205 (25) | 1.00 |
| Normal Head CT | | -5803 (256) | - |

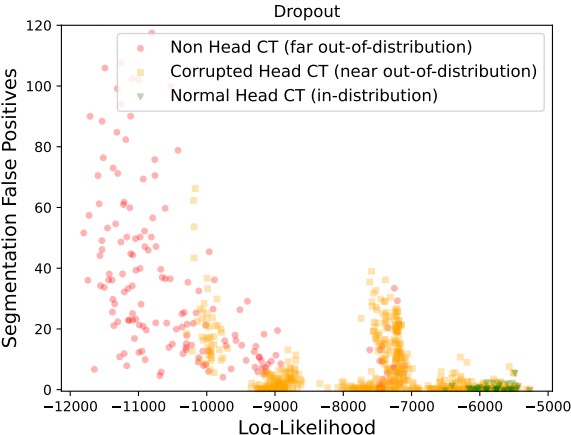

Figure 3: Image log-likelihood and the number of FP detections made by the dropout network.

## Acknowledgments

MG, PW, WS, SO, PN and MJC are supported by a grant from the Wellcome Trust (WT213038/Z/18/Z). MJC and SO are also supported by the Wellcome/EPSRC Centre for Medical Engineering (WT203148/Z/16/Z), and the InnovateUK-funded London AI centre for Value-based Healthcare. PN is also supported by NIHR UCLH Biomedical Research Centre. YM is supported by a grant from the Medical Research Council (MR/T005351/1). The models in this work were trained on NVIDIA Cambridge-1, the UK's largest supercomputer, aimed at accelerating digital biology.

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

## Appendix A. Demonstration of image corruptions

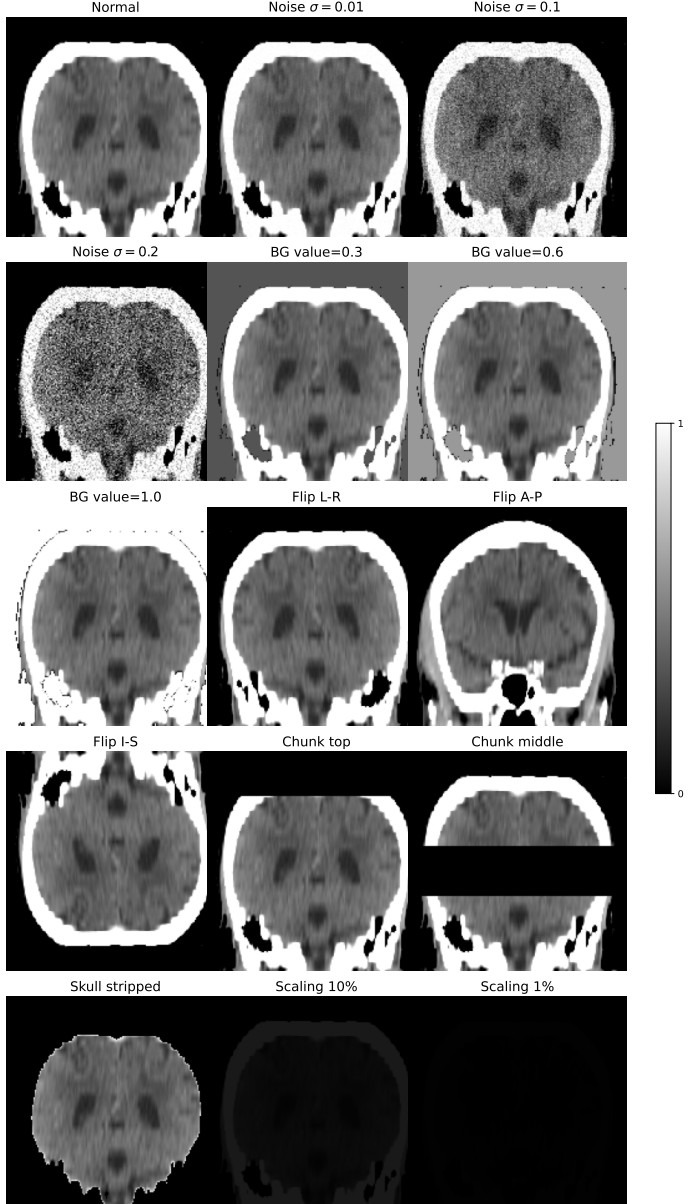

Figure A.1: Example of all the corruptions applied to one subject from the KCH head CT dataset. All images are shown with the same intensity range. Corruptions are: **Noise**: Adding Gaussian noise $\sim \mathcal{N}(0, \sigma^2)$, **BG value**: replacing background value of 0 with a new constant value, **Flip**: invert image through described plane; **Chunk**: set a number of parallel slices=0, **Skull strip**: remove skull using method described by (Muschelli et al., 2015), **Scaling**: reduce global image intensity by multiplying by a fixed factor.

## Appendix B. Additional results

Table B.1: AUC for distinguishing TP and FP lesion detections using per-lesion confidence scores from several segmentation uncertainty methods, for near-OOD corrupted head CT data. It is noteable that the Baseline network outperforms the Deep Ensemble for data with no corruptions applied. This occurs because the baseline network makes a large number of wrong, low-confidence predictions that inflate the AUC score. This illustrates that AUC scores alone are not sufficient for assessing the effectiveness of segmentation networks. However, they suffice for making the key point here: none of the networks assessed are able to provide accurate measures of uncertainty when operating OOD.

| Network | Perturbation | | | | | | |
|---|---|---|---|---|---|---|---|
| | None | Noise | Background | Flipping | Chunks | Skullstrip | Scaling |
| Baseline | 0.84 | 0.82 | 0.38 | 0.86 | 0.87 | 0.45 | 0.33 |
| Deep Ensemble | 0.75 | 0.85 | 0.42 | 0.77 | 0.82 | 0.54 | 0.36 |
| Dropout | 0.83 | 0.87 | 0.52 | 0.86 | 0.86 | 0.58 | 0.35 |

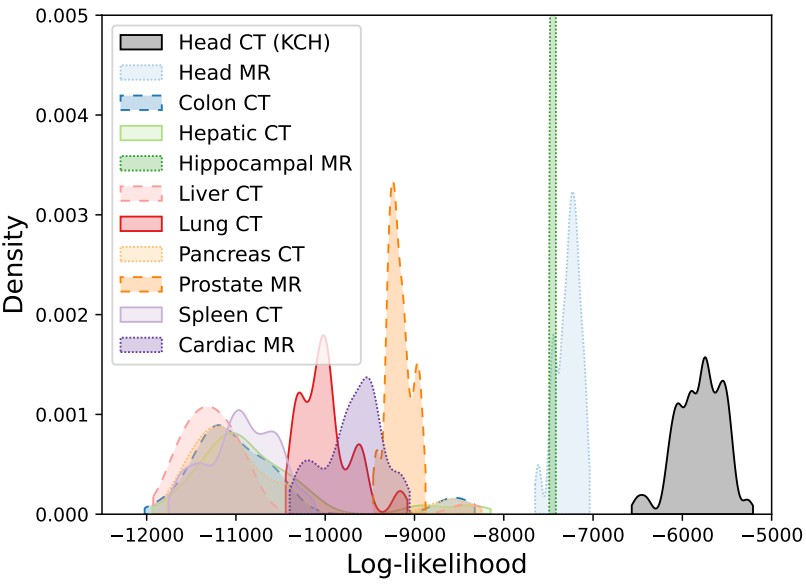

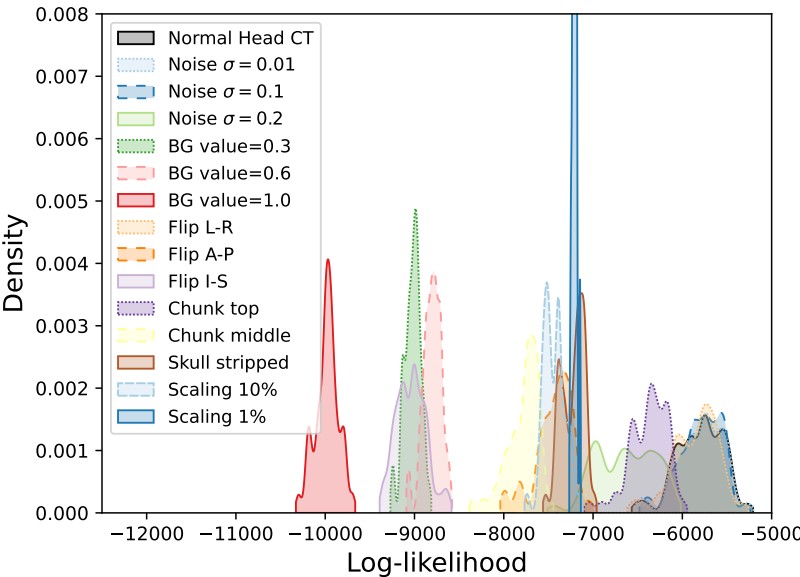

Figure B.1: Image log-likelihoods for each sub class in the far-OOD dataset (top) and near-OOD dataset (bottom).

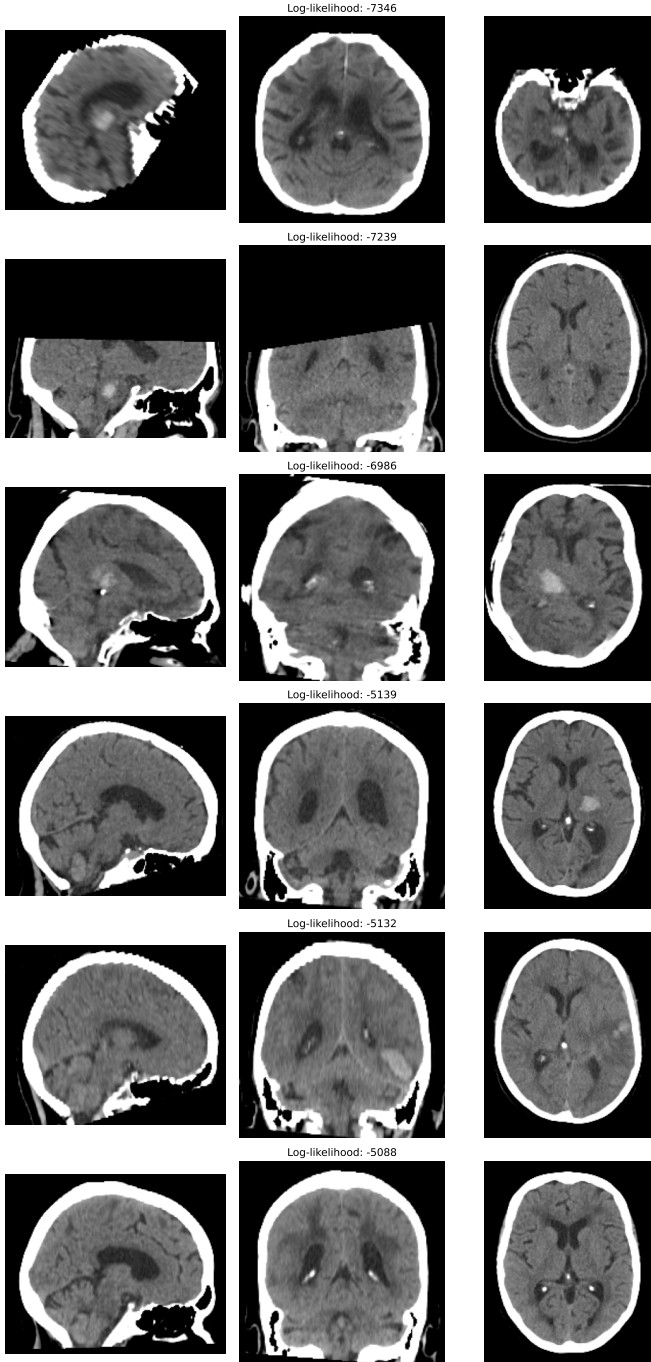

Figure B.2: For the CROMIS dataset, this shows the three volumes assigned the lowest log-likelihood values (top three rows) and the highest values (bottom three rows), with three planes shown for each volume.

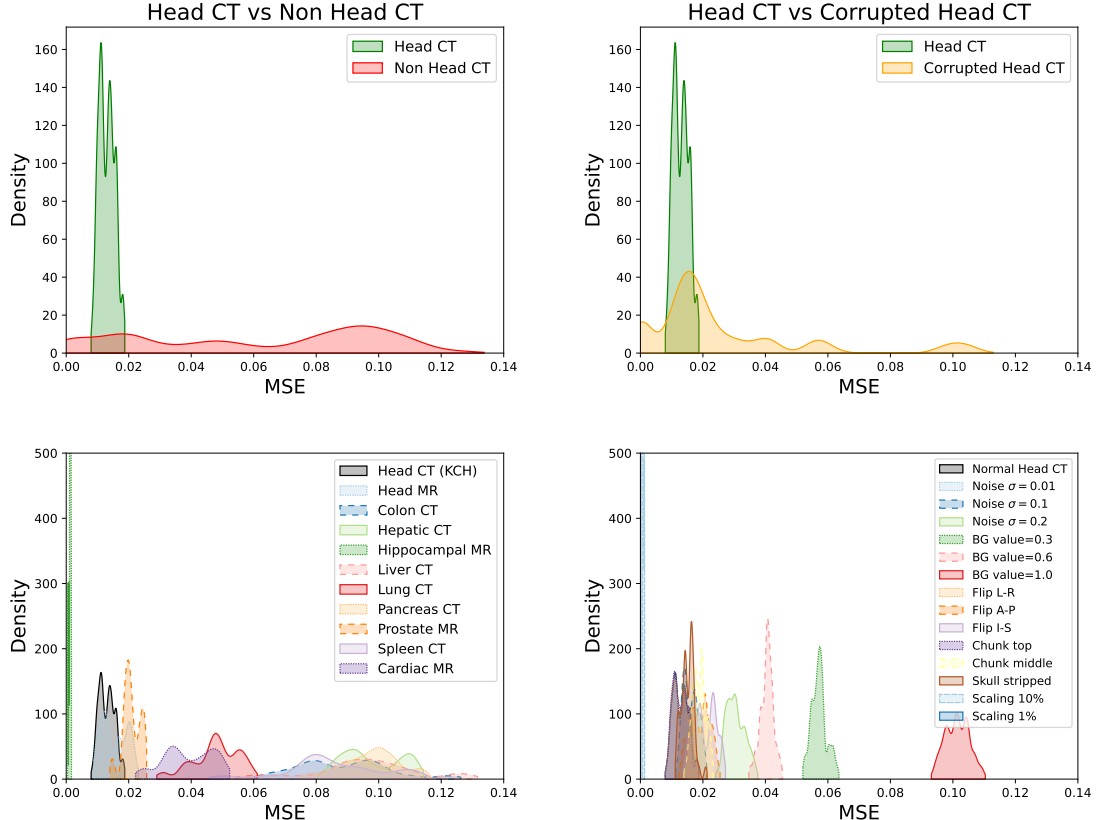

Figure B.3: VQ-GAN reconstruction mean-squared error for the far-OOD case (left column) and the near-OOD case (right column), for coarse class labels (top row) and fine-grained labels (bottom row). These results show that reconstruction MSE alone is unable to identify OOD data.

Table B.2: Study of changes to the VQ architecture and their influence on model performance. We report AUC scores for differentiating between each sub-class and normal CT data using the image log-likelihood values provided by each model (1.0 is perfect performance). Model elements changed are: **layers**: 3 or 4 layers in the encoder and decoder of the auto-encoder, **losses**: MSE - mean squared error, Perceptual - perceptual loss + MSE, Spectral - spectral+perceptual loss+MSE, **GAN**: whether or not there was an adversarial component to the training. Highlighted in red are any AUC scores $< 1.0$ for the far-OOD non head CT data, or $\leq 0.95$ for the near-OOD corrupted head CT data. An asterisk indicates the AUC score is significantly greater than the that of the baseline model (3-layer MSE, no-GAN) at a level of $p < 0.05$ as determined by bootstrapping with 1000 repetitions.

| | Dataset | Model | | | | | |
| | | 3-layer MSE no-GAN | 4-layer MSE no-GAN | 4-layer MSE GAN | 4-layer Perceptual no-GAN | 4-layer Perceptual GAN | 4-layer Spectral GAN (ours) |
|---|---|---|---|---|---|---|---|
| Non Head CT | Head MR | 0.00 | 1.00* | 0.85* | 0.84* | 1.00* | 1.00* |
| | Colon CT | 1.00 | 1.00 | 1.00 | 1.00 | 1.00 | 1.00 |
| | Hepatic CT | 1.00 | 1.00 | 1.00 | 1.00 | 1.00 | 1.00 |
| | Hippocampal MR | 0.00 | 1.00* | 0.00 | 0.01 | 1.00* | 1.00* |
| | Liver CT | 1.00 | 1.00 | 1.00 | 1.00 | 1.00 | 1.00 |
| | Lung CT | 1.00 | 1.00 | 1.00 | 1.00 | 1.00 | 1.00 |
| | Pancreas CT | 1.00 | 1.00 | 1.00 | 1.00 | 1.00 | 1.00 |
| | Prostate MR | 0.11 | 1.00* | 1.00* | 1.00* | 1.00* | 1.00* |
| | Spleen CT | 1.00 | 1.00 | 1.00 | 1.00 | 1.00 | 1.00 |
| | Cardiac MR | 1.00 | 1.00 | 1.00 | 1.00 | 1.00 | 1.00 |
| Corrupted Head CT | Noise $\sigma = 0.01$ | 0.48 | 0.49 | 0.48 | 0.49 | 0.49 | 0.49 |
| | Noise $\sigma = 0.1$ | 0.61 | 0.47 | 0.44 | 0.48 | 0.49 | 0.49 |
| | Noise $\sigma = 0.2$ | 0.82 | 0.70 | 0.71 | 0.58 | 0.80 | 0.98* |
| | BG value=0.3 | 1.00 | 1.00 | 1.00 | 1.00 | 1.00 | 1.00 |
| | BG value=0.6 | 1.00 | 1.00 | 1.00 | 1.00 | 1.00 | 1.00 |
| | BG value=1.0 | 1.00 | 1.00 | 1.00 | 1.00 | 1.00 | 1.00 |
| | Flip L-R | 0.53 | 0.55 | 0.58* | 0.55 | 0.57 | 0.55 |
| | Flip A-P | 1.00 | 1.00 | 1.00 | 1.00 | 1.00 | 1.00 |
| | Flip I-S | 1.00 | 1.00 | 1.00 | 1.00 | 1.00 | 1.00 |
| | Chunk top | 0.62 | 0.95* | 0.92* | 0.94* | 0.95* | 0.96* |
| | Chunk middle | 0.99 | 1.00 | 1.00 | 1.00 | 1.00* | 1.00 |
| | Skull stripped | 0.00 | 1.00* | 1.00* | 0.98* | 1.00* | 1.00* |
| | Scaling 10% | 0.44 | 1.00* | 0.62* | 0.85* | 1.00* | 1.00* |
| | Scaling 1% | 0.00 | 0.96* | 0.00 | 0.00 | 1.00* | 1.00* |

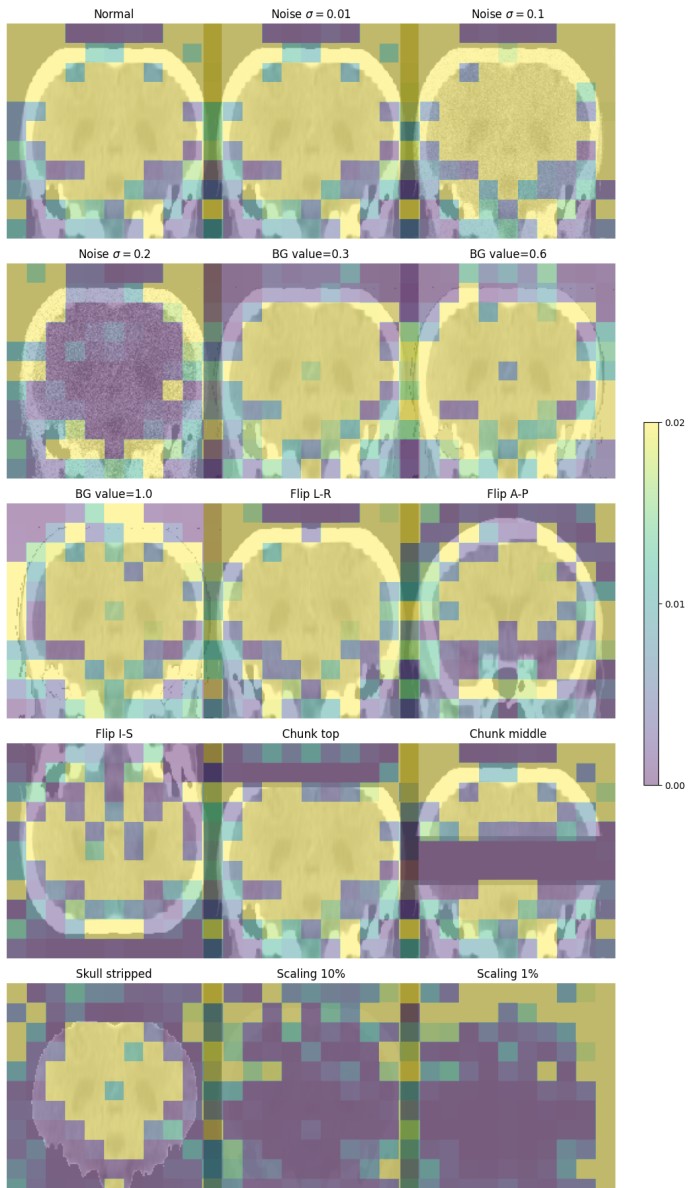

Figure B.4: Spatial likelihood maps obtained from the transformer. These are calculated by taking the likelihood values for each code in the sequence, $p(x_i|\mathbf{x}_{<i})$, reshaping from 1D back to 3D, and then upsampling by a factor of 16 along each axis using nearest-neighbour interpolation to match the size of the latent representation to the size of the input image. The network accurately assigns low likelihood values to regions of corruptions, such as the brain in the noise $\sigma = 0.2$ case, missing chunks, and the absent skull. The same images without likelihood values overlaid are shown in the appendix, Figure A.1.

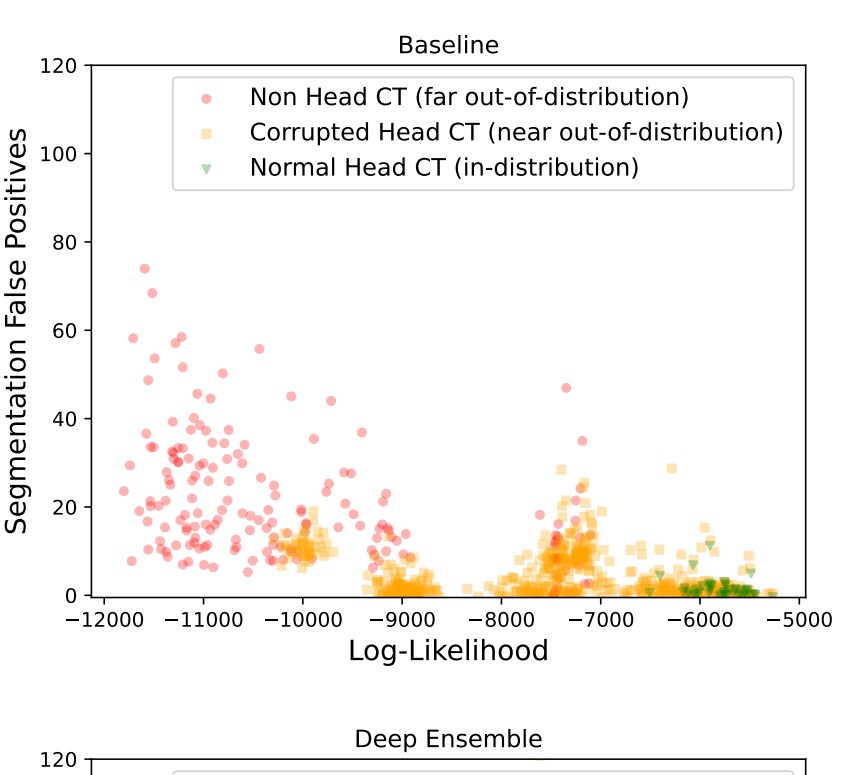

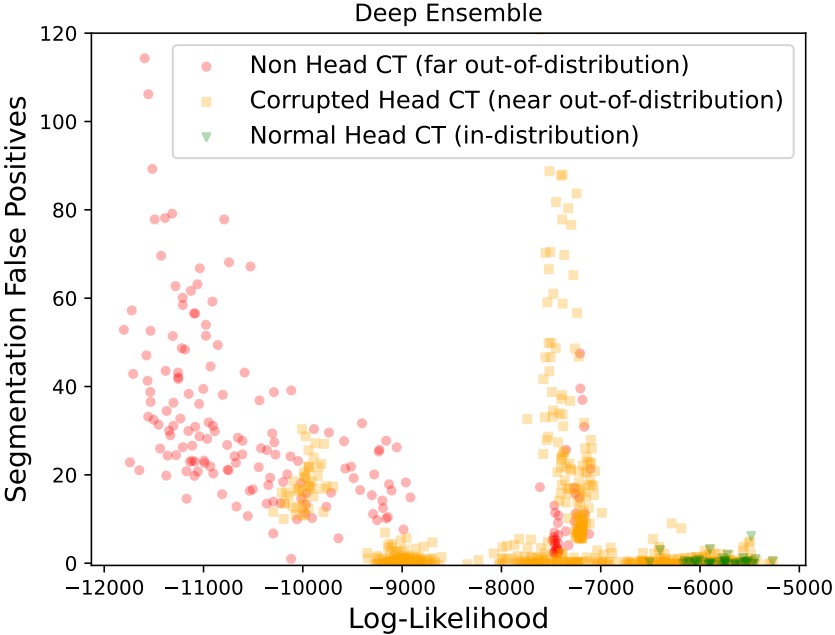

Figure B.5: Image log-likelihood and the number of FP detections.

