# OpenReview forum: "Transformer-based out-of-distribution detection for clinically safe segmentation"
_MIDL.io/2022/Conference — MIDL 2022_

### Official Review · Reviewer_4o1x · 2022-01-20

**Confidence:** 4
**Preliminary Rating:** 5
**Recommendation:** Best Paper Award, Oral

**Summary:**

This paper addresses the problem of detecting out of distribution situations with a segmentation model in mind.  Firstly, it demonstrates that a few common uncertainty quantification methods (softmax values, ensemble, MCDO) fail to recognize OOD situations reliably.  The second main contribution is the use of vector-quantized 3D GANs and transformers in order to model the likelihood of the whole 3D medical image volume to be in-distribution, which is demonstrated to be superior to the above UQ methods.  While the focus is on volume-level scores, it is also shown that the transformer can give spatial information on the specific regions that cause the input to be OOD (based on purposely introduced artifacts).  The conclusion is that the transformer-based OOD detection works well in particular in situations where the segmentation model fails.  This is a continuation of a series of MIDL submissions of the same authors based on similar technology, but with a novel application (and refined techniques).

**Strengths:**

The paper is well-written, well motivated and convincingly evaluates the benefits of the proposed approach.  I felt that the authors have given a nice summary of the state of the art and developments that led up to this work.  I particularly like the evaluation in several respects: It is based on a broad number of near- and far-OOD situations, which are summarized in the main paper and dissected a little further in the appendix. The three UQ methods are well chosen; even if they could've been optimized a little, they absolutely serve the point (to show that they might fail at OOD detection). The FP and TP definitions are lesion-based and therefore also much better for the target application than commonly used voxel-based measures.  I could follow all conclusions drawn by the authors from their results.

**Weaknesses:**

I hardly see any weakness of this paper; the obvious one is that the method is probably still too expensive to be used in many hospital settings, let alone in more resource-constrained environments.  The paper states that the models were trained on UK's largest supercomputer.  Inference will not require the same resources, of course, but this work is one of the first to apply transformers to full 3D volumes (not small ROIs) and the techniques are not easy to deploy to usual hospital settings yet. I find this relevant to be aware of and to be mentioned here, but it is not a serious weakness of the paper.

**Deanonymize Review:**

yes

**Detailed Comments:**

The plots can be improved technically in a few regards; I suggest to use a vector format (e.g., PDF instead of PNG), to make better use of the space by enlarging them (horizontally, changing the aspect ratio), and to use a fixed log likelihood range for all plots, making it easier to compare them.  (The ranges are similar enough already, so that the downsides should be negligible.)

Furthermore, the scatter plots with segmentation FPs could benefit from a logarithmic y scale?

Finally, you're the native speakers, I am not, but I found the repeated phrase "confidently wrong predictions" a little strange. Wouldn't "confidently [make/produce/...] wrong predictions" be more precise?

**Final Rating After The Rebuttal:**

5: Strong Accept

**Justification Of The Final Rating:**

I stay with my very positive rating of this paper; it is a nice contribution to the MIDL conference. Good that the figures were converted to vector graphics. The answers to my questions were good and agreeable.

**Paper Type:**

both

**Questions To Address In The Rebuttal:**

It is good that CCE loss was used for the softmax-based UQ model, but why was Dice used for the two other techniques?

Would it be possible to combine the OOD detection model with the segmentation model, and would that make sense?

Do you have any ideas for how to make the method more affordable?

("Focus on points that might change your mind." does not really apply here - I had to make questions up in order to get to 200 characters.)

**Special Issue:**

yes

---

### Official Review · Reviewer_gV8X · 2022-01-21

**Confidence:** 4
**Preliminary Rating:** 3
**Recommendation:** Oral, Poster

**Summary:**

The authors provide a method to segment hemorrhagic lesions in head CT scans using out-of-distribution detection in combination with transformer architectures. They evaluate multiple ablations in their manuscript to evaluate which network works best in combination with transformers. They have modified in-distribution scans to simulate out-of-distribution data that is then used for their evaluation.

**Strengths:**

- Spatial likelihood feature maps are a good idea. However, one can easily see/assume/verify that the flipping and all non-gaussian noise modifications have nearly no effect since the data is very, very close to the ID and thus regarded as ID instead of OOD or near-OOD.
- Table B1 for the ablation study is insightful.
- The evaluation setup and general idea are nice, especially how the authors describe their experiments and settings in terms of modifications/network settings and architecture, which makes their work easy to reproduce.


**Weaknesses:**

- The authors should clearly define the meaning of far-OOD and near-OOD since this feels like a very subjective measure. → the main difference: far is non-head CT and near are corrupted head CTs, however are non-head CTs even considered as OOD even though a whole different anatomy is used (see next point). The idea with the far-OOD is not really obvious in such a way that having a network that is trained on head CT scans and then using it on different anatomies (which would be a far-OOD as the authors describe).
- Regarding their perfect performance on the far-OOD: This is not really a huge insight, since a proper segmentation network should create an empty segmentation for a scan with a different anatomy thus one should be able to draw a hard line between non-head and head CT data log-likelihoods as it is shown in Figure 2, i.e. no overlap.
- “Furthermore, the results show that corrupted CT-scans that were assigned similar likelihoods to the in-distribution data tend to be handled better by the segmentation network, supporting our claim that these indeed are more subtle corruptions.” [p. 7] → The corrupted CT scans are defined as near-OOD. If those corruptions are only subtle, then how do the authors justify that they flag those scans as OOD since they would rather be ID because of their similarity to the ID dataset that is used ? The term near-OOD is subjective and might suggest that they are OOD and not rather ID or vice versa. The authors should specify this more clearly. → A modification can be very intense or very subtle in such a way that there is nearly no change but it would still be considered as OOD.
- The very good performance on far-OOD is somewhat expected since a different anatomy is used by the authors. A true OOD would be an experiment with the same settings but different CT machines or protocols that were followed during the image generation or different modalities but not two very different anatomies. → It is no wonder those AUC results are all 1, since there is nothing to segment.
- near-OOD: Removing the top or middle part of a scan does not make an ID scan OOD ? The same goes for flipping and even skull stripping. The gaussian noise added to a scan is more of a suitable corruption. → See images in Appendix A
- Background pixel modification as part of near-OOD: Somewhat obvious that the results are that good since the background makes up the most of those scans and changing this really significantly leads to the result that it is treated as OOD, but would this really be OOD? → I personally think that only the Gaussian noise modification from all the proposed modifications by the authors can somewhat be treated as a modification to synthetically create OOD data.
- Appendix B: Comparing the network's likelihood when trained on head CTs with Liver CT or Lung CTs does not really create any insight. More interestingly is the difference between head MRI and head CT scan which in fact might be a really ID / OOD scenario.  → Having such a good separation between head CT and Prostate MRI, Heart MRI etc. is somewhat to be expected.


**Deanonymize Review:**

no

**Detailed Comments:**

- The authors should mention how the data preprocessing part affects the quality in terms of ‘new’ artifacts and how this will impact the OOD detection in the end.
- The authors should clarify how the corruptions were made on the GT masks. Flipping and stripping is pretty obvious, but in terms of gaussian noise with a specific standard deviation, there may be a random process with a seed involved. Did they specifically set this seed equally or did they only use the same standard deviation to generate the Gaussian noise? → Did the authors implement those transformations on their own or did they use a library like TorchIO?
- The style of the charts is not that outstanding, they look very plain.
- Figure B1: the second one, ie. near-OOD: the normal head CT part (gray dashed part) is not visible
- Hippocampal MRI log-likelihood in Appendix B looks somehow weird since those are two parallel, vertical lines? → Is this just cut due to the image size or is this really the case?


**Final Rating After The Rebuttal:**

5: Strong Accept

**Justification Of The Final Rating:**

The authors thoroughly addressed the comments in a satisfying manner. The answer to why so many non-head CT scans are used is very good and should be included in the main manuscript. Further, the authors changed their images to make them more readable, as suggested. Thus, the initial score of “borderline” is raised to “strong accept” with an oral recommendation.

**Paper Type:**

both

**Questions To Address In The Rebuttal:**

- If ID data corruptions are only subtle, then how do the authors justify that they flag those scans as OOD data since they would rather be ID because of their similarity to the ID dataset that is used?
- What is the hard constraint when determining if a scan is ID or near-OOD?
- What did the authors expect when they did their experiments in terms of comparing a network's performance trained on head CT scans that is then used on Prostate MRI or Liver CT scans? Why did they use so many non-head CT examples in their results?


**Special Issue:**

no

---

### Official Review · Reviewer_QS2i · 2022-01-26

**Confidence:** 5
**Preliminary Rating:** 5
**Recommendation:** Oral

**Summary:**

This paper proposes performing a full 3D OOD detection using a VQ-GAN to estimate a latent representation. Then a transformer is used to estimate the data density distribution. In application to segmenting hemorrhages in CT, the image likelihood correlates with the segmentation quality. The method thus serves to select images with insufficient quality.

**Strengths:**

Transformers have been applied to perform OOD detection on 3D image data by estimation uncertainty through a VQ-GAN network and a transformer for data likelihood estimation.

Large datasets are included, and an extensive ablation study is performed.

The method is developed within the MONAI-framework.

**Weaknesses:**

No joint corruptions were applied to the data.

The authors use scans that were not head CT and corrupted head CT scans as OOD examples. From a clinical perspective it is more interesting to find a set of poorly segmented ICH scans and try to assess if those are OOD.

No clinical examples of detected low-quality images are given.

**Deanonymize Review:**

no

**Final Rating After The Rebuttal:**

5: Strong Accept

**Justification Of The Final Rating:**

Thank you for responding to my questions. You clearly chose for single corruptions. I acknowledge that the experiments conducted are constrained within the dataset at hand, and the available labeling data. This addresses my questions with regard to this paper.

**Paper Type:**

both

**Questions To Address In The Rebuttal:**

It seems that single corruptions were applied to the KCH dataset. Please elaborate on the effect of joint corruptions and their effect on OOD estimation. Preferably, add an additional experiment that combines (easy OOD) corruptions.

The authors use scans that were not head CT and corrupted head CT scans as OOD examples. From a clinical perspective it is more interesting to find a set of poorly segmented ICH scans and try to assess if those are OOD; are the segmentations wrong due to the data? Could the authors elaborate on how this could be studied and if the proposed method would suffice for this. Preferably, experiments should be added that assess the degree of OOD of poor segmentation examples.

The authors stated the following in the conclusion: "These results suggest our approach can be used to filter out OOD inputs that a segmentation network is likely to fail on, ensuring it is onlyrun on in-distribution data they are likely to perform well on."
I tend to disagree. In the provided study a framework is presented that could be used to study if OOD CT scans can be detected that have poor automated segmentation performance. The authors do not provide results that actually verify this. In this study only artificial OOD detection is studied and not true OOD in patient populations compared to a segmentation networks training data. I would assume that those true OOD head CTs with hemorrhage are far more difficult to detect than the easier corrupted scans. Please elaborate in the discussion.


**Special Issue:**

yes

---

### Official Review · Reviewer_VL2P · 2022-01-27

**Confidence:** 4
**Preliminary Rating:** 5
**Recommendation:** Oral

**Summary:**

The paper presents a method for unsupervised out-of-distribution detection by training a VQ-GAN on head CT scans and fitting a transformer to the resulting latent representations. By evaluating the predictability of these representations, a measure of distance to the training distribution is estimated. The method is evaluated using both non-head CT scans and corrupted head CT scans; while not all corruptions could be identified, the authors demonstrate perfect results on the set of non-head CT scans.

**Strengths:**

The writing of the paper is clear and the the structure of the paper makes it pleasant to read. The problem is well motivated, the method description is mostly clear and the experimental results of the OOD detection are well presented.

The paper presents an interesting twist on the work that introduced the VQ-GAN, which used a transformer to propose discrete tokens for its decoder. Using this transformer to evaluate the predictability of tokens is quite an elegant approach to OOD detection.

**Weaknesses:**

The authors state that the results from their ablation study demonstrate that the VQ-GAN outperforms the VQ-VAE, but these improvements seem very small. Are these differences statistically significant?

**Deanonymize Review:**

yes

**Detailed Comments:**

It is not clear to me why the transformer is trained to learn the distribution of p(x i | **x** <i). In Esser et al. 2021, this is necessary to formulate image generation as a next-index prediction, but that is not the case in this work. Would it not make more sense to learn the distribution of p(x i | **x** ≠i) ?

I was a little disappointed to see that the method was not compared to a baseline of simply evaluating the reconstruction error of the trained VQ-GAN on each image. The proposed method clearly works, but it would be valuable to disentangle how much of the heavy lifting is done by the VQ-GAN and what the added value of the transformer actually is. This could be visualised by plotting the reconstruction MSE like in Figure 2.

Why did you use the cross-entropy loss only for the baseline, rather than for all three segmentation networks? Would the improved calibration not be useful in all three settings?

The ensemble being much worse than the baseline for the non-corrupted images is quite an unintuitive result; I assume this happens because the ensemble makes fewer mistakes in the "easy" cases, which would mean the TP/FP classification problem is actually harder in this setting. It might be good to clarify how these results can or cannot be compared and how these differences should be interpreted.

While the figures are generally clear, the text in Figures 1, 2 and B.1 is too small to read comfortably on a printed version of the paper and the colours in B.1 are very difficult to differentiate. It would be helpful to add striped patterns to some of these.

**Final Rating After The Rebuttal:**

5: Strong Accept

**Justification Of The Final Rating:**

I am satisfied with the responses to my comments; after reading the updated manuscript, I am substantially more convinced of the value of the presented method. I am confident the paper will be a valuable addition to the MIDL program.

**Paper Type:**

both

**Questions To Address In The Rebuttal:**

I would like the authors to comment on the statistical significance of the differences found in the ablation study, as well as the first three points in the detailed comments. In particular I wonder about the choice of optimisation goal for the transformer.

**Special Issue:**

yes

---

### Meta-Review · Area_Chair_mQYB · 2022-02-14

**Recommendation:** Accept (Oral)
**Confidence:** 5

**Metareview:**

The work has been reviewed by four reviewers who are unanimous in their strong accepts. The paper presents a novel approach to out of distribution detection in medical imaging. The work is well-prepared and contains interesting experiments. The authors have thoroughly addressed the comments of the reviewers.

---

### Decision · Program_Chairs · 2022-02-28

Accept